# Outcome monitoring and risk stratification after cardiac procedure in neonates, infants, children and young adults born with congenital heart disease: protocol for a multicentre prospective cohort study (Children OMACp)

Mai Baquedano [1,2] Samantha E de Jesus,[1] Filippo Rapetto,[2] Gavin J Murphy,[3] Gianni Angelini [2,4] Umberto Benedetto,[2] Patricia Caldas,[2] Prashant K Srivastava,[5] Orhan Uzun,[6] Karen Luyt,[4,7] Cecilia Gonzalez Corcia,[2] Demetris Taliotis,[2] Serban Stoica,[8] Deborah A Lawlor,[4,9] Andrew R Bamber,[4,10] Alison Perry,[2] Katie L Skeffington,[11] Ikenna Omeje,[12] John Pappachan,[13] Andrew D Mumford,[14] Richard J M Coward,[1] Damien Kenny,[15] Massimo Caputo[1,11]

**Correspondence to**
Mai Baquedano;
mai.baquedano@bristol.ac.uk

## ABSTRACT

**Introduction** Congenital heart disease (CHD) represents the most common birth defect, affecting from 0.4% to 1.2% of children born in developed countries. The survival of these patients has increased significantly, but CHD remains one of the major causes of neonatal and childhood death. The aetiology of CHD is complex, with some evidence of both genetic and environmental causes. However, there is still lack of knowledge regarding modifiable risk factors and molecular and genetic mechanisms underlying the development of CHD. This study aims to develop a prospective cohort of patients undergoing cardiac procedures that will bring together routinely collected clinical data and biological samples from patients and their biological mothers, in order to investigate risk factors and predictors of postoperative-outcomes, as well as better understanding the effect of the surgical intervention on the early and long-term outcomes.

**Methods and analysis** Children OMACp (OMACp, outcome monitoring after cardiac procedure in congenital heart disease) is a multicentre, prospective cohort study recruiting children with CHD undergoing a cardiac procedure. The study aims to recruit 3000 participants over 5 years (2019–2024) across multiple UK sites. Routine clinical data will be collected, as well as participant questionnaires collecting sociodemographic, NHS resource use and quality of life data. Biological samples (blood, urine and surgical waste tissue from patients, and blood and urine samples from biological mothers) will be collected where consent has been obtained. Follow-up outcome and questionnaire data will be collected for 5 years.

**Ethics and dissemination** The study was approved by the London–Brent Research Ethics Committee on 30

## STRENGTHS AND LIMITATIONS OF THIS STUDY

⇒ Large multicentre, prospective cohort study collecting clinical/questionnaire data and biosamples from children with congenital heart disease (CHD) undergoing cardiac procedure and their biological mothers.

⇒ Combined data and samples analyses will allow identification of risk factors and short-term, medium-term and long-term outcome predictors.

⇒ Cost-effective method of generating a persistent data and sample registry accessible both nationally and internationally.

⇒ The cohort does not include control data from healthy subjects; however, this will be obtained from external sources and record linkage.

⇒ Only patients requiring a cardiac procedure are included, not representing all CHDs.

July 2019 (19/SW/0113). Participants (or their parent/guardian if under 16 years of age) must provide informed consent prior to being recruited into the study. Mothers who wish to take part must also provide informed consent prior to being recruited. The study is sponsored by University Hospitals Bristol and Weston Foundation Trust and is managed by the University of Bristol. Children OMACp is adopted onto the National Institute for Health Research Clinical Research Network portfolio. Findings will be disseminated through peer-reviewed publications, presentation at conference, meetings and through patient organisations and newsletters.

**Trial registration number** ISRCTN17650644.

## INTRODUCTION

Congenital heart disease (CHD) represents the most common birth defect, affecting from 0.4% to 1.2% of live-born children in the developed countries,[1 2] and remains one of the major causes of neonatal, infant and childhood death.[2 3] CHD is generally defined as a gross structural abnormality of the heart or intrathoracic great vessels present at birth that is actually or potentially of functional significance.[4]

The aetiology of CHD is complex, with some evidence of both genetic and environmental causes. However, there is still a lack of knowledge regarding modifiable causal risk factors.[5] Furthermore, the molecular and genetic mechanisms underlying the development of CHD have not been clearly elucidated and the identification of specific subphenotypes of CHD remains challenging.[6 7]

While the incidence of CHD has remained unchanged, the survival of these neonates and infants has increased dramatically in the last two decades,[8] leading to a rise in the numbers of adults with CHD. For example, in 2000, there were nearly equal numbers of adults and children with severe CHD,[9] highlighting the need to better understand CHD aetiology, including modifiable risk factors and molecular genetic mechanisms, so that methods of preventing CHD can be developed to reduce its disease burden.[10]

Current research on potentially modifiable maternal risk factors suggests that maternal smoking, greater adiposity and associated dyslipidaemia, gestational diabetes and hypertensive disorders of pregnancy are associated with increased incidence of CHD.[11–13] However, most of these studies have been cross-sectional or case control, with relatively small number of cases or have not adjusted for key confounders and therefore cannot determine degree of causality. Genome-wide association studies (GWAS) of the fetal/child genome including up to 2000 cases have identified some robust genetic variants associated with all or specific CHDs, but lack power to identify modest genetic associations, particularly with subtypes of CHD.[14] GWAS of maternal genotype are rare and few aetiological studies have used methods to try and unpick causal effects from association, such as using genetic variants as instrumental variables in Mendelian randomisation studies, negative control studies or within sibship analyses.[15] It is necessary to apply best methods to establish the most relevant targets for interventions in order to facilitate future randomised controlled trials assessing causality.

Patients with CHD often require multiple complex surgeries and life-long medical therapies and follow-up. These therapies are ineffective in up to 50% of the treated patients and involve significant morbidity, long-term mortality and substantial cost.[16]

The development of predictive tools and integrated information for advancing CHD clinical diagnosis and prognosis is one of the most prominent challenges of clinical cardiology. Traditional research methods have largely failed to advance an understanding of early complications and longer-term failure of cardiac surgery interventions. However, application of large-scale genomic[17] and metabolomic[18 19] approaches to the prediction of complications and applied epidemiological techniques to investigate causality[20 21] have the potential to advance mechanistic understanding and streamline therapeutic development.[22]

In parallel, Electronic Health Records data generated and collected during normal clinical care are increasingly being linked and used for translational cardiovascular research.

Our ability to advance medical care and efficiently translate science into modern medicine is bounded by our capacity to access and process these 'Big Data'. From human genetics and 'omics' to routine clinical documentation and from internal imaging to motion capture, the volume and variety of 'Big Data' is bound to change in medicine. Statistical, mathematical, visualisation and computational approaches, from a wide range of disciplines, as well as systems for innovative interventions, are needed to keep apace of the complexity in disease prediction and prognostics and to advance medicine.

The study aims to develop a prospective cohort of patients undergoing surgery and/or catheterisation that will bring together routinely collected clinical data as well as biological samples from both patients and their biological mothers, in order to investigate further risk factors of CHD and predictors of postoperative outcomes. Our ambition is to make this a persistent resource accessible to high-quality future CHD studies, nationally and internationally. Outcomes and the corresponding data and biomarkers to be examined will be directed by the hypothesis of substudies and/or external ethically approved studies accessing the registry.

## METHODS AND ANALYSIS
### Study design

Children OMACp (OMACp, outcome monitoring after cardiac procedure in congenital heart disease) is a multicentre, prospective cohort study recruiting patients diagnosed with CHD and undergoing a cardiac procedure (catheterisation and/or surgery) and their biological mothers. The study aims to evaluate short-term, medium-term and long-term health outcomes and collect biological samples for patients undergoing cardiac surgery and/or catheterisation for CHD. Biological samples and risk factors information will also be collected from patients' biological mothers where consent has been obtained.

### Study setting

The cohort will recruit patients attending Bristol Royal Hospital for Children, and up to 11 additional hospitals. At the time of writing, the study is recruiting at Bristol Royal Hospital for Children, Leicester Royal Infirmary and Dublin Children's Hospital.

The planned duration of the study is 10 years (5 years recruitment period and 5 years for follow-up).

## Research questions

Specific research questions and objectives include, but are not limited to:

1. To identify the maternal genetic and potentially modifiable risk factors for CHD.
2. To identify clinical, genetic and environmental factors that act as predictors for short-term, medium-term and long-term clinical outcomes.
3. To determine the effect of clinical management strategies on outcomes of CHD including perioperative and operative care.
4. To identify factors that influence National Health Service (NHS) resource use for patients with CHD.
5. To evaluate the impact of CHD on neurological development and school performance.
6. To evaluate the impact of CHD surgical correction on organ function (such as heart, liver, kidney that are potentially damaged during the operation).
7. To explore the potential of deep phenotyping extreme high-risk groups including those with single ventricle and cyanotic heart disease.
8. To identify the impact of CHD on sleep disorders.
9. To evaluate the accuracy and prediction of imaging cardiac data with early and long-term complications.
10. To use machine learning platforms for developing personalised treatment for CHD (eg, pharmacogenetics and epigenetics).

## Study population

The target population includes neonates, infants and children (0–15 years) and young adults (16–18 years) with CHD undergoing catheterisation and/or surgery at participating centres, and their biological mothers. The study aims to recruit 3000 participants over 5 years (2019–2024).

### Inclusion criteria

Patients must:

1. be aged between 0 months and 18 years of age.
2. diagnosed with CHD.
3. undergoing cardiac surgery and/or catheterisation.
   The person giving consent must:
4. have capacity to consent (if participant 16–18 years of age) or assent (if participant age 11–15).
5. have capacity to consent and parental responsibility for the participant (if participant under 16 years of age).
   Mothers must:
1. have a biological child enrolled in Children OMACp.
2. have capacity to consent.

### Exclusion criteria

The study will exclude patients and biological mothers who are unable to give informed consent and/or assent, those with a main residence outside of the UK and Ireland and those under the care of Social Services.

## Research procedures

### Identification of participants and consent

Recruitment will be from September 2019 to August 2024. The majority of patients will be identified by the clinical team from the cardiac surgical/catheter waiting lists. Some patients may be identified on admission, 12–24 hours prior to surgery. All eligible patients, and their biological mother, will be invited to participate in the study. The study is conducted in accordance with the Declaration of Helsinki. All biological samples and questionnaires are optional.

### Recruitment

Before admission for cardiac surgery and/or catheterisation, participants and/or their parent/guardian, as well as their biological mother when possible, will be provided with participant information leaflets. These contain a study description including study details, study contacts and details of withdrawal processes. Following a period of at least 12 hours to consider the study, participants who are interested in joining will be asked to provide informed consent. Participants (if 16 or over) or their parent/guardian (if participant age is 0–15) will provide consent to routine data collection. Participants and/or parents/guardian can also optionally consent to biological samples being taken, stored and analysed, as well as to complete study questionnaires at baseline and during follow-up. Patients who are aged 11–15 years at the time of consent will also be asked to complete an assent form. This assent form will be respected. Participants who turn 16 years old during the study's follow-up period will also be asked to sign a consent form to confirm if they are happy to continue taking part in the study. Biological mothers will provide consent for their own data and samples, if they so wish. Consent will be collected electronically where possible.

### Samples and questionnaires

Several biosamples and questionnaire data can be optionally obtained from patients, and their biological mothers, as part of the study protocol.

### Patient samples

Table 1 summarises the samples that will be collected.

For patients undergoing catheterisation, blood and urine samples will be taken at baseline only, before their procedure. For patients undergoing cardiac surgery, blood and urine samples will be taken preoperatively, and postoperatively, on arrival to PICU and at 24 hours after surgery. All blood samples will be taken through lines that are already in place as part of routine care and blood sample volumes will vary depending on participant age/weight from 1 to 4 mL. Total research blood volumes drawn will take into account routinely collected bloods to ensure that patient safety is maintained. Participating centres may also agree to collect optional blood samples for RNA/DNA analyses (using PaxGene RNA tubes).

**Table 1** Participant samples

| Surgical patients | Preop | On arrival to PICU | 24 hours postarrival to PICU |
|---|---|---|---|
| Blood | Yes | Yes | Yes |
| Urine | Yes | Yes | Yes |
| PaxGene RNA | Yes | Yes | Yes |
| Waste tissue | Yes | N/A | N/A |
| Catheter patients | | | |
| Blood | Yes | N/A | N/A |
| Urine | Yes | N/A | N/A |
| PaxGene RNA | Yes | N/A | N/A |
| Waste tissue | N/A | N/A | N/A |

N/A, not applicable; PICU, paediatric intensive care unit.

**Table 2** Mothers' samples

| Mothers | Baseline |
|---|---|
| Blood | Yes |
| Urine | Yes |
| PaxGene RNA | Yes |

Waste cardiac tissue and fluids that would otherwise be discarded will also be collected from surgical patients, where consent has been obtained.

Blood and urine samples will be transferred to the local laboratory for processing. Samples will then be stored at −80°C. Waste tissue samples will be snap frozen in liquid nitrogen and stored at −80°C, unless analyses of fresh tissue samples are required for a specific substudy, in which case, samples will be processed before freezing.

### Patient questionnaires and follow-up data

If the patients are 8 years old or older, a quality-of-life questionnaire that combines the KIDSCREEN (SCREENing for and Promotion of Health Related Quality of Life in Children an Adolescents), CHU9D (Child Health Utility) and EQ5DY (EuroQol 5 dimension youth) will also be completed by the patient or their parent/guardian prior to the procedure (or within a few days postprocedure), at 3 months and 12 month postprocedure, and every year thereafter for up to 5 years.

All patients, or parents on their behalf if under 16 years, will also be asked to complete a general health and resource use questionnaire at 12 months postprocedure and every year thereafter for up to 5 years.

In addition to the yearly questionnaires, the patient's medical notes will be reviewed yearly for 5 years post-procedure.

### Mothers' samples and data

Consented mothers will donate blood and urine samples and will complete a baseline questionnaire to provide demographic and risk factor data. Prenatal medical notes will be accessed where possible.

Blood and urine samples will be transferred to the local laboratory for processing. Samples will then be stored at −80°C. Table 2 provides a summary of maternal samples to be collected.

### Data collection

Data will be collected from patients' medical notes, hospital databases and hospital episodes statistics and ethically approved questionnaires. Data will include, but is not limited to demographic, lifestyle and socioeconomic data, as well as maternal risk factors, intraoperative details, postoperative complications, NHS resource use and quality of life. Results from the analysis of biological samples (eg, genomics, metabolomics, proteomics and transcriptomics) will also form part of the final dataset.

All data will be handled in line with the General Data Protection Regulations and Good Clinical Practice Guidelines. Data will be collected electronically on specifically designed case report forms and entered onto a REDCap database. Data will be stored on secured servers and access to the database will be restricted to authorised staff only.

### Analysis

The study will provide a resource for several research questions to be addressed (see above). There are two broad types of analyses that the cohort will be used for: (a) analyses that addresses questions requiring data from patients with CHD only and (b) analyses addressing questions that require a healthy control/comparison group.

A key focus initially will be on prediction and risk stratification, in particular developing accurate tools for predicting those children who are at most risk of adverse outcomes after surgery, and who are therefore likely to benefit from tailored care-packages and more intense monitoring post-surgery.

Sociodemographic, clinical and 'omics' (eg, genomics, proteomics, epigenomics and metabolomics) data will be considered for inclusion in the prediction models. Machine learning approaches, that allow for a large number of potential predictors to be studied simultaneously, while accounting for their correlation structure and potential interactions to be taken into account, will be used to identify a 'top-set' of potential prediction models. We will then consider those models further with respect to discrimination, calibration and clinical utility, selecting those that maximise cost-effectiveness. We will use internal validation procedures but also (with extension of the network beyond Bristol) have the ability to externally validate the prediction tool in cohorts from outside of Bristol.

With addition of maternal data to the study, we will also explore causal maternal risk factors using multivariable regression analyses. We will also contribute data and academic input to GWAS consortia to identify both fetal

and maternal genetic variants associated with CHD and use those in two-sample Mendelian randomization.

When reporting outcomes, participants will be grouped according to the type of operation and type of CHD. The Children OMACp cohort allows for the collection of routine clinical operative and perioperative data alongside short (eg, acute kidney injury) and long-term (eg, mortality) clinical outcomes. We will use data and biosamples to answer hypothesis-driven research questions such as assessing the effect of cardiopulmonary bypass duration on renal injury. Analyses will be appropriately powered assessing subgroups of the overall cohort for individual research questions.

When relevant, we will explore options for comparison/control groups, including data from studies such as ALSPAC (Avon Longitudinal Study Of Pregnancy And Childhood), as well as data linkage. They will provide suitable healthy controls to explore genetic and environmental determinants of CHD.

### Patient and public involvement

A patient and public involvement (PPI) group reviewed the study documents before the initial ethics application was submitted and comments were incorporated as appropriate.

Our unit has also been leading many PPI initiative within the OMACp project. Our work in this context focuses on: (a) conceiving public presentations of collaborative work created with partners including artists, for broad public engagement and raising awareness around CHD; (b) devise and hold workshops for engagement with patient audience exploring their narratives of illness; (c) developing creative collaborations across and beyond the University, including academic and community partners; (d) participating in outreach events and (e) studying the dynamics of arts-and-health collaboration.

### Withdrawal

All participants will be informed of their right to withdraw from the study at any time, without needing to give a reason. Any data and samples already collected about the participant will remain for analysis unless otherwise requested by the participant.

## ETHICS AND DISSEMINATION
### Ethics

The study was approved by the London–Brent Research Ethics Committee on 30 July 2019 (19/SW/0113) and is adopted onto the National Institute for Health Research Clinical Research Network portfolio. The study is sponsored by University Hospitals Bristol and Weston Foundation Trust and is managed by the University of Bristol and has organised indemnification. Protocol deviations will be documented and reported to the chief investigator and the sponsor immediately.

Given this is an observational study that does not change the patient's standard care, there are no risks to patient safety resulting from the study. Child blood samples are taken through pre-existing lines inserted for clinical indications within safe volumes for child weight. Collecting urine and waste tissue samples does not pose any additional risks. Therefore, it is not possible for clinical adverse events to be attributed to study-specific procedures. For mothers, there is a very minimal risk associated with blood withdrawal. Participants (or their parent/guardian if under 16 years of age) must provide informed consent prior to being recruited into the study. Mothers who wish to take part must also provide informed consent prior to being recruited.

### Dissemination

Baseline characteristics of the cohort and any finding will be presented at local, national and international meetings and published in peer-reviewed papers. Findings will also be presented to patients through patient organisations and newsletters.

The resource generated from this study, which includes routine data, samples and questionnaire data, will be made available to other researchers for future studies nationally and internationally. Our aim is to establish this as a persistent resource that will be made accessible to high-quality, ethically approved research by submitting a proposal and access request to be considered by the Children OMACp Executive Group, similar to procedures used by general cohorts such as UK Biobank, ALSPAC and CRANE (The Cleft Registry and Audit NEtwork). A full description of the data and samples available will be accessible through the study's website.

### Changes to protocol since ethical approval

The study began recruiting in September 2019 using version 2 of the protocol. When the study began, it was single-centre, recruiting only in Bristol, and patients only completed quality of life questionnaires at 3 months and 12 months postprocedure. In January 2020, the study became multicentre and in January 2021 yearly general health and quality of life questionnaires were introduced for the 5 year follow-up period. The current version of the protocol is version 5 dated 1 March 2022.

## DISCUSSION

Here we present the Children OMACp study, a unique resource facilitating research into CHD by enabling efficient collection of large amounts of data from medical records and questionnaires, as well as biological samples from multiple hospitals performing heart procedures on this specific cohort of patients and their biological mothers.

OMACp has many strengths, providing the ability to develop a cohort of patients with CHD in UK and Ireland, for robust hypothesis-driven research and will focus initially on identification of risk factors and outcome predictors for short-term, medium-term and long-term outcomes, offering the basis to explore risk, prognosis,

outcome and management. The final goal is to create a cost-effective, persistent, data and biosample repository that can be made available to other high-quality ethically approved studies to support future CHD research nationally and internationally.

Combining perioperative data, maternal risk factors information and biosamples, the study will explore the potential for machine learning and personalised medicine, by utilising artificial intelligence models to review sociodemographic, clinical (including imaging data) and multiple 'omics (eg, genomic, epigenomic, proteomics and metabolomics) to better risk stratified the extreme large heterogeneous diagnosis of CHD and their surgical early-term and long-term outcomes and prepare individually tailored approaches to care. The collection of maternal data will contribute information on the potential causality of maternal risk factors.

While initially, the study will collect only minimal information on father's history (via the mother's questionnaire), in future, the aim is to extend the recruitment to fathers to assess impact of father's genetics, medical and lifestyle factors on children CHD risk and outcomes. Furthermore, there will be potential for including additional non-cardiac malformations.

Although only patients requiring a cardiac procedure are included, representing a fraction of all CHDs, this is the group that will most greatly benefit from any advances in the management and tailoring of medical care.

In addition, while the cohort does not include healthy subjects, the study has access to use data and samples from external sources (eg, ALSPAC and record linkage) as healthy controls, when necessary.

This study will further explore the potential of deep phenotyping extreme high-risk groups, including those with single ventricle and cyanotic heart disease, as well as evaluating the long-term impact of CHD treatment on neurocognitive development

**Author affiliations**
[1]Translational Health Sciences, University of Bristol, Bristol, UK
[2]University Hospitals Bristol and Weston NHS Foundation Trust, Bristol, UK
[3]Department of Cardiovascular Sciences and NIHR Leicester Biomedical Research Unit in Cardiovascular Medicine, NIHR Leicester Biomedical Research Centre Cardiovascular Diseases, Leicester, UK
[4]Bristol Medical School, University of Bristol, Bristol, UK
[5]Imperial College London, London, UK
[6]Cardiff and Vale NHS Trust, Cardiff, UK
[7]NICU, University Hospitals Bristol and Weston NHS Foundation Trust, Bristol, UK
[8]Bristol Royal Hospital for Children, Bristol, UK
[9]MRC Integrative Epidemiology Unit, University of Bristol, Bristol, UK
[10]North Bristol NHS Trust, Westbury on Trym, UK
[11]Bristol Heart Institute, University of Bristol, Bristol, UK
[12]University Hospitals of Leicester NHS Trust, Leicester, UK
[13]PICU, Southampton Children's Hospital, Southampton, UK
[14]Department of Haematology, University Hospitals Bristol and Weston NHS Foundation Trust, Bristol, UK
[15]Children's Health Ireland at Crumlin, Dublin, Crumlin, Ireland

**Acknowledgements** We are grateful to the patients and families who supported our patient involvement work and to the patients and families that have consented to and already provided data to the study.

**Contributors** MC and MB conceptualised the project. MB, MC, DL and UB developed the protocol. FR, GM, GA, UB, PC, PKS, OU, KL, CGC, DT, SS, DL, AB, AP, KLS, IO, JP, AM, RJMC and DK approved protocol design. MB and SEdJ contributed to the oversight and conduct of the study. MB designed and managed the data collection systems. MC, GM, DL and IO contributed to the funding of the study. SEdJ and MB drafted the study protocol manuscript and received comments from all coauthors. All authors have read and approved the final manuscript.

**Funding** The study is funded by the NIHR Bristol Biomedical Research Centre (BRC-1215-20011) and Professor Massimo Caputo's British Heart Foundation Personal Chairs (BHF-CH/17/1/32804). Deborah Lawlor's contribution to the study is supported by her British Heart Foundation Chair (CH/F/20/90003) and the MRC Integrative Epidemiology Unit (MC_UU_00011/6). Gavin Murphy's contribution to the study is supported by the BHF (CH/12/1/29419 and RG/17/9/32812). Ikenna Omeje's contribution to the study is supported by the Heart Link Children's charity.

**Competing interests** DL has received support from Medtronic and Roche Diagnostics for research unrelated to this publication.

**Patient and public involvement** Patients and/or the public were involved in the design, or conduct, or reporting or dissemination plans of this research. Refer to the Methods section for further details.

**Patient consent for publication** Not required.

**Provenance and peer review** Not commissioned; externally peer reviewed.

**ORCID iDs**
Mai Baquedano http://orcid.org/0000-0002-7101-3082
Gianni Angelini http://orcid.org/0000-0002-1753-3730

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
