## [Reviewer comments · BMJ Open]

ARTICLE DETAILS

TITLE (PROVISIONAL)	Outcome monitoring and risk stratification after cardiac procedure in neonates, infants, children and young adults born with congenital heart disease: protocol for a multi-centre prospective cohort study (Children OMACp)
AUTHORS	Baquedano, Mai; de Jesus, Samantha; Rapetto, Filippo; Murphy, Gavin; Angelini, Gianni; Benedetto, Umberto; Caldas, Patricia; Srivastava, Prashant K.; Uzun, Orhan; Luyt, Karen; Gonzalez Corcia, Cecilia; Taliotis, Demetris; Stoica, Serban; Lawlor, Deborah; Bamber, Andrew; Perry, Alison; Skeffington, Katie L.; Omeje, Ikenna; Pappachan, John; Mumford, Andrew; Coward, Richard J.M.; Kenny, Damien; Caputo, Massimo

VERSION 1 – REVIEW

REVIEWER	Andrea Giulio Quarti Azienda Ospedaliero-Universitaria di Bologna IRCCS, Pediatric Cardiac Surgery and Adult Congenital Heart Disease Program
REVIEW RETURNED	16-Apr-2023

GENERAL COMMENTS	To the author: The authors describe a multicenter prospective cohort study that is recruiting patients with CHD requiring cardiac surgery or a cath-lab procedure. This is a very interesting study that could finally answer many unanswered questions about CHD from a large patient cohort. The study design and setting are clear. The research questions are also clear. Also, I couldn't find any ethics issues. I have very few questions for the authors: Aims of the study are to evaluate risk factors influencing early/medium and long-term outcome; to highlight potentially modifiable maternal risk factors; to evaluate the impact of CHD and therapeutic approaches on the NHS. -I would like to get more details on what data will be collected: Could the authors describe what data will be examined from blood and urine and waste tissue? Do they want to create some sort of sample bank or are there specific details that have not been discussed in this document? - I could understand that collecting samples even from the father has many possible practical consequences that are difficult to overcome, however the lack of details about the father is a potential source of bias. The father's genetics, for example, are a potential source of CHD, although their weight in determining CHD is less than the mother's genetics. Father who smokes increases the incidence of CHD. If the father's genetics present a problem for the study, in my opinion the father's health history could be of interest. -Does the author believe that postoperative blood sampling (especially for genetics) upon arrival at the PICU could represent a bias at least in small newborns? In this cohort of patients, the CPB is
--

	primed with donor blood which could potentially alter the genetics of the sample. -Collecting data from CHD requiring surgery or a Cath-Lab procedure will narrow down to CHD requiring treatment. that represent only a fraction of all CHDs. This should be within the limits of the study. Overall, I really like this protocol and believe it will add a lot of details in understanding CHD and their optimal treatment.
--	--

REVIEWER	Nicolas Joram CHU Nantes
REVIEW RETURNED	09-May-2023

GENERAL COMMENTS	Dear authors and editors, Thank you for giving me the opportunity to review this protocol paper. The manuscript reports an ongoing multi-centre, prospective, cohort study protocol. The aim of this study is to investigate the risk factors and predictors of post-operative outcomes among neonates, infants, children and young adults born with congenital heart disease and undergoing cardiac procedure. The rationale, methods, analysis, ethics concerns and dissemination of the finding planned in the study are clearly detailed. The analysis of the cohort will focus initially on risk stratification for short, medium and long-term outcomes. The final goal of this cohort will be to create a database available for future CHD research. The authors mention the main limitation that is the lack of inclusion control data from healthy subjects. However, they mention the ability to use data from other studies in order to provide healthy controls. This cohort study is of great clinical interest and the paper is very clear and well written. The planned duration of the study is clearly stated. Nine research questions are described. I have very few remarks and queries: Major comments:  - Globally, the study aims to address a broad range of research questions. Then, many collected biological samples and clinical/questionnaire data are mentioned but no clear outcome or judgment criterion appears in the manuscript. In particular, short-, medium- and long-term outcomes, even without details, should be more clearly exposed. - In the abstract, the authors mention the aim of understanding the efficacy of the surgical intervention. This point is not specifically presented in the list of research questions or in the rest of the manuscript. - In the list of research questions, one of them is to investigate the optimal clinical management. This point should be specified and, again, the mention of judgment criteria would be interesting. - In the summary, the authors mention the exploration of deep phenotyping the extreme high-risk group of patients with single ventricle and cyanotic heart disease. This point is not mentioned before in the manuscript, especially in the list of research questions. Minor comment : In page 13, line 13, the authors mention a reference #23 which does not appear in the list the paragraph "References".
--

REVIEWER	Shazia Mohsin Sindh Institute of Urology and Transplantation, Paediatrics and Child Health
REVIEW RETURNED	09-May-2023

GENERAL COMMENTS	this is a very well written protocol. i must congratulate authors in coming up with this collaborative registry. i am sure after collection of data this registry will bring meaningful contribution to CHD cohort
--

VERSION 1 – AUTHOR RESPONSE

	Reviewer 1 - Dr. Andrea Giulio Quarti, Azienda Ospedaliero-Universitaria di Bologna IRCCS	
1.	I would like to get more details on what data will be collected: Could the authors describe what data will be examined from blood and urine and waste tissue? Do they want to create some sort of sample bank or are there specific details that have not been discussed in this document?	This study will create a large data and sample registry accessible to sub-studies and other ethically approved studies. Outcomes and the corresponding data and biomarkers to be examined will be directed by the hypothesis of sub-studies and/or external ethically approved studies accessing the registry. For example, as part of a sub-study looking at interleukins 1 and 6 as predictors of risk of low cardiac output syndrome (LCOS), we have analysed interleukins 1 and 6 in a subset of blood samples and compared with clinical LCOS diagnoses.
2.	I could understand that collecting samples even from the father has many possible practical consequences that are difficult to overcome, however the lack of details about the father is a potential source of bias. The father's genetics, for example, are a potential source of CHD, although their weight in determining CHD is less than the mother's genetics. Father who smokes increases the incidence of CHD. If the father's genetics present a problem for the study, in my opinion the father's health history could be of interest.	The authors agree with this statement and have added it as an additional limitation. No samples and only minimal data collected from fathers as resource intensive. Initially we decided to focus on study feasibility and collect only minimal information on father's history (via the mother's questionnaire) to reduce impact on limited resources. In future, we were planning to recruit fathers, and collect data and samples from them, to assess impact of father's genetics, medical and lifestyle factors on children CHD risk and outcomes.

		We were also planning to include additional malformations, not just CHD. To this purpose we have designed and started recruitment for a more recent study – PEARL (Surgical-PEARL protocol: a multicentre prospective cohort study exploring aetiology, management and outcomes for patients with congenital anomalies potentially requiring surgical intervention BMJ Open) which is an extension of OMACP and recruits patients and both their mothers and fathers.
3.	Does the author believe that postoperative blood sampling (especially for genetics) upon arrival at the PICU could represent a bias at least in small newborns? In this cohort of patients, the CPB is primed with donor blood which could potentially alter the genetics of the sample.	The authors agree that this is a potential limitation of genetic analyses in this patient cohort. Data on pre-operative weight, donor blood and transfusions are collected routinely in the perioperative period. We will exclude these patients from any genetic analyses which may be confounded by donor blood. However, other time points (e.g. pre operative and 24hrs post PICU arrival) may be used for analysis.
4.	Collecting data from CHD requiring surgery or a Cath-Lab procedure will narrow down to CHD requiring treatment. that represent only a fraction of all CHDs. This should be within the limits of the study	The authors agreed this is an important comment and have added this to the study limitations section of the paper: Only patients requiring a cardiac procedure are included, not representing all CHDs.
	Reviewer 2 - Dr. Nicolas Joram, CHU Nantes	
1.	Globally, the study aims to address a broad range of research questions. Then, many collected biological samples and clinical/questionnaire data are mentioned but no clear outcome or judgment criterion appears in	This study will create a large data and sample registry. Outcomes and the corresponding data and biomarkers to be examined will be

	the manuscript. In particular, short-, medium- and long-term outcomes, even without details, should be more clearly exposed.	directed by the hypothesis of sub-studies and/or external ethically approved studies accessing the registry. For example, as part of a sub-study looking at identifying patients at risk of low cardiac output syndrome (LCOS), we have analysed interleukins 1 and 6 in a subset of the samples and compared with clinical LCOS diagnoses. The following ahs been added to the end of the Introduction in the paper: Outcomes and the corresponding data and biomarkers to be examined will be directed by the hypothesis of sub-studies and/or external ethically approved studies accessing the registry.
2.	In the abstract, the authors mention the aim of understanding the efficacy of the surgical intervention. This point is not specifically presented in the list of research questions or in the rest of the manuscript.	Changed in abstract to 'effect' of surgical intervention. Altered research questions to reflect this: c 'To determine the optimal clinical management of CHD including the effect of peri-operative and operative care on outcomes.
3.	In the list of research questions, one of them is to investigate the optimal clinical management. This point should be specified and, again, the mention of judgment criteria would be interesting.	'Optimal' removed as the authors agree it is challenging to define optimal in this context. Reworded to: c To determine the effect of clinical management strategies on outcomes of CHD including peri-operative and operative care. Added section below to analyses section: The children OMACp cohort allows for the collection of routine clinical operative and peri-operative data alongside short (e.g. acute kidney

		injury) and long term (e.g. mortality) clinical outcomes. We will utilise data and biosamples to answer hypothesis driven research questions such as assessing the effect of CPB duration on renal injury.
4.	In the summary, the authors mention the exploration of deep phenotyping the extreme high-risk group of patients with single ventricle and cyanotic heart disease. This point is not mentioned before in the manuscript, especially in the list of research questions	This was an oversight and has been added to the research questions. To explore the potential of deep phenotyping extreme high-risk groups including those with single ventricle and cyanotic heart disease
5.	In page 13, line 13, the authors mention a reference #23 which does not appear in the list the paragraph "References".	Reference added in error. Deleted.
Reviewer 3 - Dr. Shazia Mohsin, Sindh Institute of Urology and Transplantation		
	No revisions proposed or queries raised	

VERSION 2 – REVIEW

REVIEWER	Andrea Giulio Quarti Azienda Ospedaliero-Universitaria di Bologna IRCCS, Pediatric Cardiac Surgery and Adult Congenital Heart Disease Program
REVIEW RETURNED	21-Jul-2023

GENERAL COMMENTS	To the author: thank you for answering my questions. I have no further revisions to propose.
--